# Deep learning-based image enhancement in optical coherence tomography by exploiting interference fringe

Woojin Lee[1], Hyeong Soo Nam[1], Jae Yeon Seok[2], Wang-Yuhl Oh [1], Jin Won Kim [3] & Hongki Yoo [1✉]

Optical coherence tomography (OCT), an interferometric imaging technique, provides non-invasive, high-speed, high-sensitive volumetric biological imaging in vivo. However, systemic features inherent in the basic operating principle of OCT limit its imaging performance such as spatial resolution and signal-to-noise ratio. Here, we propose a deep learning-based OCT image enhancement framework that exploits raw interference fringes to achieve further enhancement from currently obtainable optimized images. The proposed framework for enhancing spatial resolution and reducing speckle noise in OCT images consists of two separate models: an A-scan-based network (NetA) and a B-scan-based network (NetB). NetA utilizes spectrograms obtained via short-time Fourier transform of raw interference fringes to enhance axial resolution of A-scans. NetB was introduced to enhance lateral resolution and reduce speckle noise in B-scan images. The individually trained networks were applied sequentially. We demonstrate the versatility and capability of the proposed framework by visually and quantitatively validating its robust performance. Comparative studies suggest that deep learning utilizing interference fringes can outperform the existing methods. Furthermore, we demonstrate the advantages of the proposed method by comparing our outcomes with multi-B-scan averaged images and contrast-adjusted images. We expect that the proposed framework will be a versatile technology that can improve functionality of OCT.

[1] Department of Mechanical Engineering, Korea Advanced Institute of Science and Technology, 291 Daehak-ro, Yuseong-gu, Daejeon 34141, Republic of Korea. [2] Department of Pathology, Yongin Severance Hospital, Yonsei University College of Medicine, 363 Dongbaekjukjeon-daero, Giheung-gu, Yongin-si, Gyeonggi-do 16995, Republic of Korea. [3] Multimodal Imaging and Theranostic Lab, Cardiovascular Center, Korea University Guro Hospital, 148 Gurodong-ro, Guro-gu, Seoul 08308, Republic of Korea. ✉email: h.yoo@kaist.ac.kr

Optical coherence tomography (OCT) is an indispensable optical imaging modality that can provide non-invasive three-dimensional imaging in vivo with high-speed and high-sensitivity[1]. OCT operates based on an interferometric technique that uses coherent detection of backscattered light from sample and reference arms using a broad-band light source[2]. The difference in the optical length of each arm is encoded in the frequency domain of the detected interference signal. Therefore, the depth profile of a sample, commonly referred to as a A-scan, is typically retrieved by applying Fourier transform to the measured interference signal. The laser beam is then scanned laterally to obtain a two-dimensional cross-sectional OCT image with a depth-lateral axis, commonly referred to as a B-scan. Based on these fundamental principles, OCT with microscopic spatial resolution is widely used as a diagnostic tool in various medical fields, such as ophthalmology and cardiology[3,4]. However, OCT applications often suffer from systemic limitations arising from the basic operating principle; these limitations include the presence of speckle noise, limited depth-of-focus (DOF), and degradation in spatial resolution. In detail, speckle noise deteriorates detailed morphological information by reducing contrast of OCT images[5]. The axial resolution is physically determined by the spectral bandwidth of the light source, while the lateral resolution is dominated mainly by the numerical aperture of the imaging optics[6]. The lateral resolution is also only maintained within a limited DOF, reducing the effective imaging range[7]. In addition, OCT exploiting a broadband light source requires the development of sophisticated optical systems[2,7–9]. Therefore, to fully utilize the diagnostic potential of OCT in many preclinical and clinical applications, it is of great importance to enhance OCT images by overcoming these drawbacks.

While hardware-based enhancement techniques require expensive high-performance lasers and/or additional optical components to improve OCT images[8–11], software-based approaches can achieve such enhancement with only minimal modification to the underlying system. Conventional software-based studies have proposed spectral estimation[12] and spectrum-shaping[13] for enhancing resolution, and B-scan averaging[14] and filtering-based methods[15–17] for suppressing noise. However, these methods often require relatively time-consuming iteration algorithms for the enhancement, introducing spurious artifacts and oversmoothing the images. Meanwhile, as an alternative to overcome the limitations of conventional methods, the enhancement of OCT images through deep learning, which has recently shown outstanding performances in numerous fields, is drawing attention. Recent researches have shown that deep learning can outperform handcrafted feature descriptors in a number of imaging processing fields[18–23]. Inspired by these studies, deep learning is being applied actively to various optical imaging modalities, including OCT, for super resolution and noise reduction[24–29]. Several models have been reported that can restore the resolution of intentionally degraded OCT images[30–32]. Elsewhere, deep learning methods have been proposed to reduce speckle by learning frame-averaged OCT images[33,34] or speckle-modulating OCT[35] as ground truths. Liang, et al.[36] implemented conditional generative adversarial networks (GAN) to enhance spatial resolution while preserving detailed speckle patterns in OCT images.

However, previously-proposed deep learning-based OCT image processing methods have used only grayscale 8-bit OCT images. Considering that OCT A-scans are constructed by applying signal processing steps including a Fourier transform to the interference fringes, valuable information might be lost during Fourier transformation, log compression, and conversion to 8-bit. Therefore, better image enhancement may be achieved if the raw interference fringe is fully utilized. However, in general, commercial OCT provides only gray-scale images; even with custom-built OCT, raw interference fringes in the wavenumber (k) domain have never been fully exploited to enhance OCT image quality based on deep learning. In addition, many studies have shown how accurately degraded inputs can be reconstructed to the ground truth through the trained models, and no further enhancement has been shown in the state-of-the-art images. Therefore, OCT image enhancement methods that can further enhance the current optimal image quality is needed to improve versatility and expandability by addressing the aforementioned limitations.

In this study, we propose a deep learning-based framework to enhance the quality of currently optimized OCT images. Our model consists of two separate models, an A-scan-based network (NetA) and B-scan-based network (NetB). In particular, we fully exploit the information of the raw interference fringe signal, which was partially lost during transformation to OCT images by conventional processing. NetA is mainly responsible for enhancing the axial resolution of A-scans by utilizing spectrograms, which are obtained via short-time Fourier transform (STFT) of raw interference fringes. NetB was designed to enhance lateral resolution and reduce speckle noise in OCT B-scans. The dual models were individually trained and then sequentially applied in the inference phase. The performance was also evaluated using datasets acquired from different OCT systems, thereby demonstrating the versatility and expandability of the proposed technique. The performance of this dual model deep learning-based processing was evaluated through comparative studies to other methods on the same dataset. Advantages were also demonstrated through comparisons with mutli-B-scan averaged images and contrast-adjusted images. By overcoming the aforementioned limitations of conventional OCT image processing, the performance of the proposed deep learning-based OCT signal processing framework suggests that it can be a promising technology to enhance OCT images and expand OCT functionality.

## Results

An overall schematic including training and inferences for the proposed deep learning-based OCT image enhancement framework is presented in Fig. 1. The dual model, composed of NetA and NetB, is designed based on GAN[37], consisting of generators and discriminators. NetA, which mainly enhances axial resolution, directly receives two adjacent fringes and processes them by transforming through STFT and FFT to generate spectrograms and typical OCT A-scans, respectively (Fig. 1a). While the acquired interference fringes contain depth information in the spectral domain (i.e., the k-domain), OCT A-scans, Fourier transforms of the interference fringes, lose depth-dependent spectral information with changes in k. Since the interference fringes are inevitably affected by multiple scattering (i.e., sources of speckle noise), spectral dependency of sample, and dispersion, which are highly dependent on the k-domain, the spectral information according to the change in k can be better utilized for OCT image reconstruction. Therefore, to more delicately process this invaluable information embedded in the interference fringe, a spectrogram, which is the STFT result of the fringe, was provided to the proposed deep learning-based framework as input data. Furthermore, since the spectrograms are two-dimensional (depth and $k(t)$), they can be processed like two-dimensional images, making them suitable for processing with convolution-based deep learning methods. On the other hand, NetB is introduced to enhance lateral resolution and reduce speckle noise by receiving B-scan images (Fig. 1b). Note that NetB receives a log-compressed FFT amplitude spectrum with a single-precision floating point. Typically, OCT images are presented in 8-bit

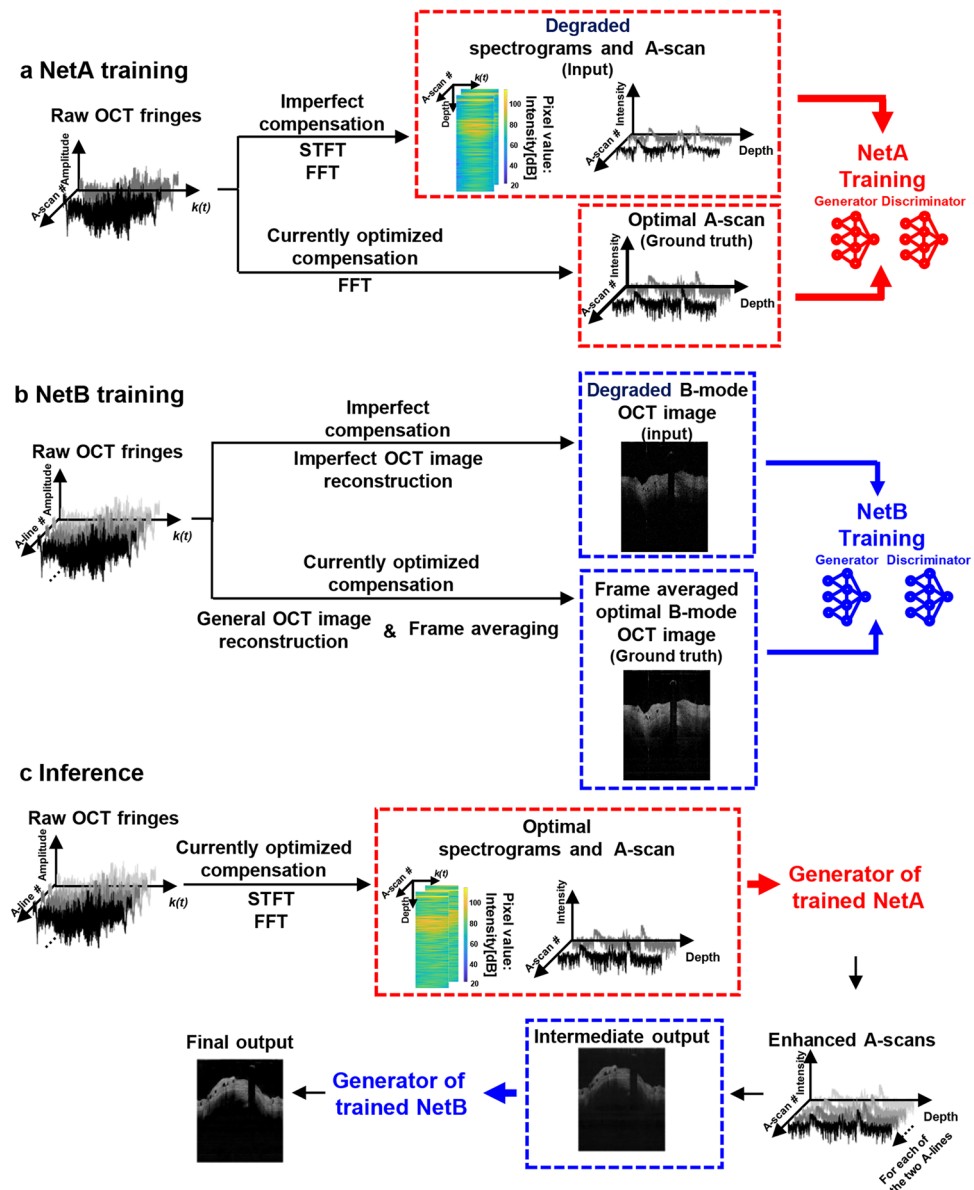

**Fig. 1 Schematic of deep learning-based OCT image enhancement framework. a** Training phase of NetA. Based on A-scans, NetA takes two adjacent interference fringes and uses them as input after transforming them into spectrograms and typical OCT A-scans. **b** Training phase of NetB. NetB is implemented based on OCT B-scans consisting of 1024 A-scans. In both networks, the ground truth was constructed utilizing the best currently available data through numerical post-processing. In NetB, the average of 7 adjacent B-scans was used as the ground truth. Inputs were intentionally degraded spectrograms, A-scans, and B-scans that have been processed with degradations such as imperfect dispersion compensation, bandwidth truncation, and SNR deterioration. **c** Inference phase of proposed OCT image enhancement framework. Data are processed using only the generators of the two trained networks. Note that currently optimized interference fringes are used as input to further enhance state-of-the-art OCT images. NetA processes A-scans sequentially for one frame and NetB processes results of Net A to produce final output.

grayscale with a limited contrast range on the dB scale, resulting in loss of information outside of the specified contrast range. Therefore, NetB receives all amplitudes of single precision floating point, without contrast limit, allowing all meaningful features to be considered without sacrificing. Optimal A-scans, acquired by applying currently optimized compensation and FFT, were used as the ground truth, while degraded spectrograms and A-scans were used as input for NetA (Fig. 1a). Degraded B-scan OCT images were used as input, and frame-averaged images of 7 adjacent optimal B-scan OCT images were used as ground truth for NetB (Fig. 1b). Note that the total interval of the 7 OCT images was specified at the lateral resolution level of the OCT system to achieve adequate noise reduction while avoiding

excessive spatial smoothing. After individual training, only NetA and NetB generators were sequentially applied to the raw OCT fringes to produce the final enhanced imaging output (Fig. 1c). The training dataset was constructed using a customized benchtop-based swept-source OCT (SS-OCT)[38] system from a variety of samples including thyroid tissue specimens, finger nails, fingertips, cucumbers, grapes, lemons, pork meat, and Scotch tape. Additional data not referenced during training were also acquired with the same OCT system to demonstrate expandability. Furthermore, in vivo data from swine coronary artery and rabbit abdominal aorta, obtained using a customized catheter-based SS-OCT system[4,39], were also used to support more robust expandability of the proposed method. Details on

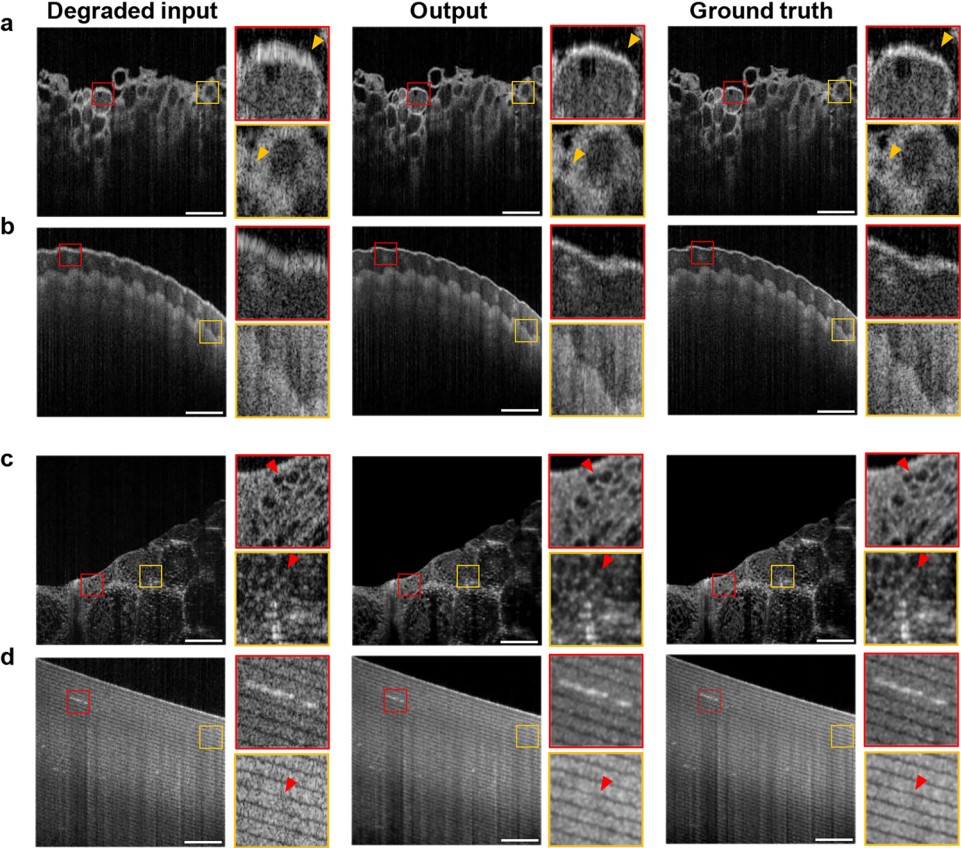

**Fig. 2 Individual test performance of both models on degraded input data.** Example result of NetA with samples of **a** thyroid tissue and **b** finger. Example result of NetB with samples of **c** lemon and **d** Scotch tape. The degraded input, output, and ground truth images are shown in the three columns. The ROIs (red and yellow boxes) on the right side of the image show magnified views (5X). Scale bars, 1 mm.

implementation and dataset can be found in the "Materials and methods" section.

**Performance evaluation of individual models.** After successful training, as indicated by training loss curves (Supplementary Fig. 1), the performance of each trained model was individually evaluated by comparing how similar each model's inference for the degraded input was to the ground truth. Note that the evaluation is based on a test set that is not referenced at all in the training. The inferenced examples for each model are shown in Fig. 2. In Fig. 2a, b, degraded input was generated by applying FFT to the degraded fringes; the output was reconstructed by individually applying NetA to all degraded fringes consisting of single B-scan images. Compared with the degraded input, the output represents a well-reconstructed structure close to the ground truth; it was obtained by typical OCT image reconstruction with optimal compensation. In particular, it is clearly manifested in the morphological features estimated as typical follicle structure of a normal thyroid tissue (yellow arrowheads in Fig. 2a). Figure 2c, d shows NetB performance on OCT images of lemon and Scotch tape. NetB output suppresses speckle noise, distinct from the degraded input, and exhibits a homogenized intensity distribution within the sample (red arrowheads in Fig. 2c, d).

Using mean square error (MSE), the structural similarity index (SSIM)[40], and the multi-scale SSIM (MS-SSIM)[41], which can measure the similarity between two images (see Supplementary Note 1 for details), the performance of each model was quantitatively evaluated for data randomly selected from the test

**Table 1 Comparison of MSE, SSIM, and MS-SSIM of input and output for NetA and NetB.**

|  | Degraded input | | Output | |
|---|---|---|---|---|
|  | **Average** | **Std** | **Average** | **Std** |
| NetA |  |  |  |  |
| MSE | 34.26 | 14.62 | 14.00 | 2.83 |
| SSIM | 0.544 | 0.138 | 0.706 | 0.102 |
| MS-SSIM | 0.767 | 0.105 | 0.910 | 0.072 |
| NetB |  |  |  |  |
| MSE | 19.290 | 11.609 | 4.078 | 1.079 |
| SSIM | 0.258 | 0.072 | 0.767 | 0.104 |
| MS-SSIM | 0.472 | 0.067 | 0.938 | 0.071 |

set. All metrics were calculated based on the ground truth. The evaluation results are summarized in Table 1. In both models, the MSE results were lower in the output than in the degraded input. Furthermore, the SSIM and the MS-SSIM figures, which were high in the output for both models, indicate that the structural similarity in the output is much akin to the ground truth at both local and global scales. Overall, outputs of models achieved results much closer to the ground truth than did the degraded input. As can be seen from the above results, both trained models were able to successfully reconstruct the degraded input and make it similar to the ground truth.

**Performance evaluation of entire framework to enhance OCT images.** The purpose of this study-was to further improve

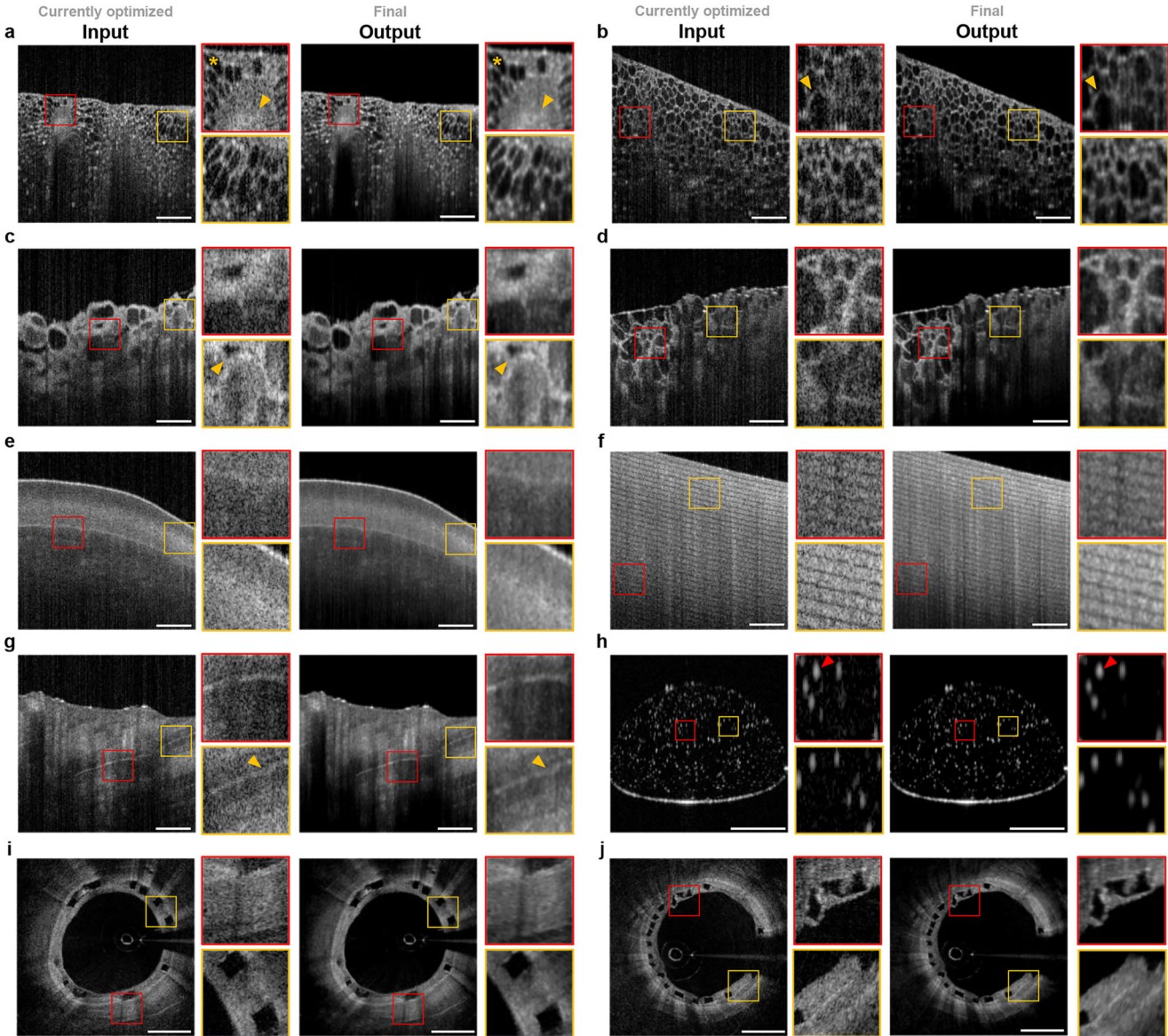

**Fig. 3 Blind testing performance of deep learning-based OCT image enhancement framework.** Left and right columns represent currently optimized input and enhanced final output, respectively. Note that, in these results, the currently optimized OCT data were the input. Final output is the result of sequentially applying both NetA and NetB to the input. Each figure represents **a** cucumber, **b** grape, **c**, **d** thyroid tissue specimen, **e** finger nail, **f** Scotch tape, **g** pork meat, **h** droplet with TiO2 microspheres, and **i**, **j** arterial cross-section. The ROIs (red and yellow boxes) on the right side of the image show magnified views (3X for **a–g** and **i–j**; 5X for **h**). Scale bars, 1 mm.

currently available optimal OCT images; so, unlike the afore-mentioned individual model evaluations, ground truth-level data were fed as input to the generators of the two trained models. As can be seen in the overall schematic in Fig. 1c, the raw fringes of OCT were fed into NetA, and the output of NetA went straight into NetB. The results of NetA for the input were denoted as intermediate output; the results of NetB were denoted as final output (Fig. 1c). All subsequent evaluation results are described based on the final output. Figure 3 shows representative OCT images of the currently optimized input and the enhanced final output. In Fig. 3a, an example of a cucumber cross-section shows that the speckle noise appearing within the tissue (arrowheads in Fig. 3a) was reduced, while visual representation of structural features such as parenchyma (asterisks in Fig. 3a) improved. Such improvement also appears in all of the samples referenced for training shown in Fig. 3b–g. Figure 3h–j shows examples of two other types of samples (microspheres and arterial cross-sections).

Note that the data for these samples were used only for perfor-mance evaluation, not for training. In particular, the system used to image the arterial tissue was different from the one used to acquire the training data set. Since these samples have completely different structural features from those in the training set, the results can demonstrate the robust reliability and expandability of the proposed deep learning-based framework. Results of micro-spheres show enhanced spatial resolution, especially axial reso-lution, indicating significantly smaller bead sizes (red arrowheads in Fig. 3h). Furthermore, the results for the arterial cross-section, shown in Fig. 3i, j, confirm that our processing can achieve robust performance for biological samples obtained from other systems. The quantitative evaluation for these results is summarized in Table 2. Since there are no clear answers to the deep learning results, we used several parameters as performance indicators, including peak signal-to-noise ratio (PSNR), the beta parameter ($\beta$), and the edge preservation factor (EPF), which can measure

**Table 2 Quantitative performance evaluation results of deep learning-based OCT image enhancement framework for reducing noise (PSNR), preserving morphological features (Beta parameter and edge preservation factor), and enhancing spatial resolution (axial resolution and lateral resolution).**

|  | Currently optimized input | | Final output | |
|---|---|---|---|---|
|  | Average | Std | Average | Std |
| PSNR | – | – | 24.96 | 0.65 |
| Beta parameter | – | – | 0.923 | 0.036 |
| Edge preservation factor | – | – | 0.996 | 0.005 |
| Axial resolution [μm] | 12.39 | 2.52 | 10.08 | 2.55 |
| Lateral resolution [μm] | 14.15 | 3.99 | 12.39 | 4.10 |

the degree of improvement in image quality. These metrics are commonly used for OCT denoising studies[42–46]. PSNR was used to quantify the noise levels in improved OCT images relative to original images. $\beta$ is a normalized metric that can measure the degree of preservation of morphological features in denoised images and is often used as a performance evaluation indicator in OCT studies for speckle reduction[42,46]. EPF shows edge preservation effects with respect to original images, computed using the local correlation[43–45] (see Supplementary Note 1 for details). On average, PSNR was improved by 24.955 dB compared to the input. $\beta$ and the EPF were evaluated and found to be 0.923 and 0.996, respectively, indicating that the spatial features were well maintained. Axial and lateral resolutions, defined as the full width at 3 dB lower than the peak intensity, were measured from images of microspheres (Fig. 3h). The enhancement in resolution was quantified by ~1.2 times in both the axial and the lateral resolution. The enhancement performance according to the degree of input degradation was additionally performed to thoroughly verify the generalization performance of the proposed method (Supplementary Fig. 2). The findings indicate that the proposed method exhibits robust generalization performance for different levels of degradation, generating desirable outcomes while avoiding over-smoothing regardless of the degradation level. Therefore, these results reveal that overall image quality was noticeably improved by reducing noise and enhancing spatial resolution, while spatial features were well preserved.

We further examined the advantages of our proposed method by comparing the results to mutli-B-scan averaged images and contrast-adjusted images. Figure 4a–d shows the comparison results with the averaged average of 7 and 21 B-scans. The B-scan averaging method using adjacent OCT frames provides exceptional speckle reduction performance[47]. However, excessive multi-B-scan averaged images result in noticeable blurring of morphological feature information due to spatial averaging (red arrowheads in Fig. 4a, c). On the other hand, it can be confirmed that these morphological features are well preserved in our proposed method (red arrowheads in Fig. 4d), which are more prominent than the averaged image of 7 B-scans (red arrowheads in Fig. 4b). Even in the quantitative evaluation, our method achieved higher PNSR, β and EPF values compared to the averaged images of 7 and 21 B scans. These results reveal that our deep learning-based framework has significant speckle reduction capability while preserving spatial feature information well, while multi-B-scan averaged image reduces speckle noise at the cost of spatial blurring. In addition, spatial resolutions are substantially enhanced even when the signal is weak (see Supplementary Fig. 3 for details). Therefore, these results revealed an noticeable improvement in overall image quality with reduced noise and

enhanced spatial resolution, and spatial features were well preserved. To confirm that both models were successfully trained without overfitting, a multi-fold analysis was also performed (Supplementary Fig. 4). Of the eight datasets, one was used as a validation set and the other seven as training sets, which were used for further training and quantitative evaluation. Accordingly, all metrics showed equivalent results to those evaluated by the originally trained model, validating our model was successfully trained without overfitting.

**Comparative studies with other methods**. The potential of the proposed deep learning-based framework to other methods was presented through comparative studies using the same dataset. We compared our method with conventional methods based on statistical filtering (block-matching 3D (BM3D)[15] and K-SVD[48]) and six other previous deep learning techniques showing reliable performance in image improvement (super-resolution convolutional neural network (SRCNN)[49], super-resolution residual neural network (SRResNet)[50], Unet[51], very-deep super-resolution (VDSR)[52], cycle-consistent adversarial network (CycleGAN)[53], and paired image-to-image translation (Pix2Pix)[54]). Deep learning techniques were adopted as-is for each of the proposed implementations, but retrained using the same dataset in this study. The training loss curves of each technique are shown in Supplementary Fig. 5. Figure 5 shows comparison results using datasets of thyroid carcinoma specimen (Fig. 5a) and microspheres (Fig. 5b). These results show that our processing outperforms most of the existing methods in suppressing the noise and preserving the edge detail and spatial content compared to the input images (Fig. 5a). In contrast, other methods, especially the conventional filtering methods such as BM3D and K-SVD, provide output images with severely blurred edges due to excessive smoothness, resulting in loss of spatial information. In addition, the resolution enhancement of our method in both the axial and lateral directions is incomparable (Fig. 5b). Quantitative evaluation using the aforementioned metrics demonstrated the superiority of our method (Fig. 5c–f). Our method showed better SNR (Fig. 5c), and preserved spatial features and edges (Fig. 5d, e). The resolution enhancement was more pronounced in the resolution measurements using microspheres (Fig. 5f). Visual evaluation, as well as quantitative comparison, show that our processing enabled effective noise reduction within tissue while preserving spatial feature information. These results suggest that only our processing may further enhance currently optimized OCT images in terms of both spatial resolution and SNR.

## Discussion
In this study, we proposed a deep learning-based OCT image processing framework to enhance spatial resolution and SNR; we then verified the performance through comparative study and spectral analysis. The PSNR was improved by 24.96 dB compared to the input, and the resolution enhancement was 1.2 times in both the axial and lateral directions. In particular, spatial features were well preserved without excessive smoothing, which was often observed in previous studies.

Importantly, we hypothesized that utilizing the spectral information contained in the OCT interference fringes could enhance the OCT imaging performance. Since raw interference fringes are very complex to be used directly for training deep learning networks, STFT was applied to obtain spectrograms that still contain spectral information. In addition, since the spectrograms are two-dimensional, they can be effectively used in deep learning networks that are known to perform well on images. We believe that this approach of utilizing spectrograms offers unique advantages over previous methods using only OCT images. While the

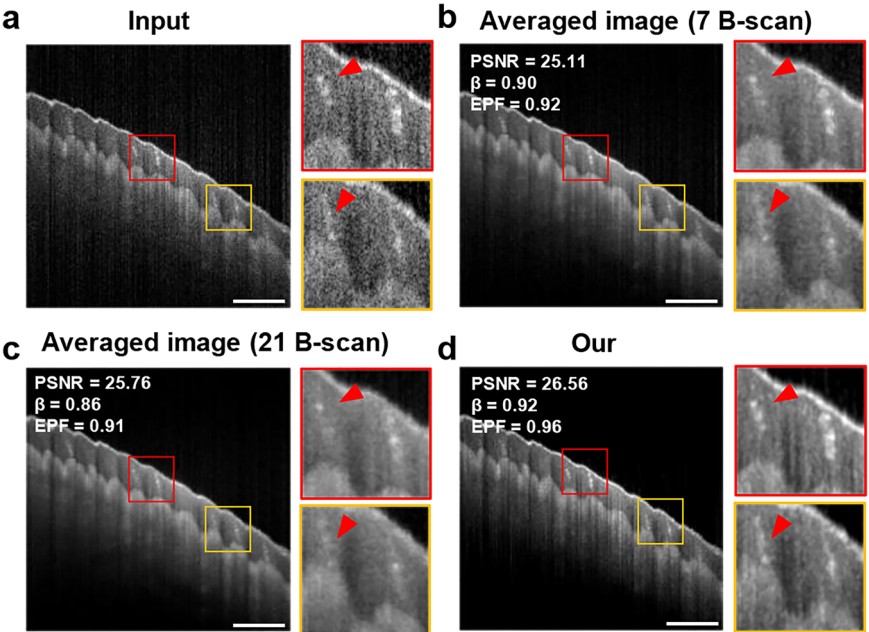

**Fig. 4 Comparison with multi-B-scan average method. a** A single B-scan of the finger tip used as input to the deep learning framework. **b**, **c** Results of speckle reduction via multi-B-scan averaging of 7 and 21 B-scans, respectively. The multi-B-scans were acquired with a frame interval of 2 μm. **d** Output processed by the proposed deep learning-based framework. The ROIs (red and yellow boxes) on the right side of the image show magnified views (3X). Scale bars, 1 mm.

spectral bandwidth directly limits the theoretical axial resolution of the OCT, we postulate that the proposed deep learning network can improve the axial resolution by restoring weak spectral information outside the spectral bandwidth of the light source.

We further investigated and verified the advantages of the proposed method by comparing the results with multi-B-scan averaged images. B-scan averaging method effectively reduces unwanted multiple-scattering-related speckle noise by averaging adjacent images in the out-of-plane direction. However, excessive B-scan averaging can inevitably cause blurring, resulting in loss of spatial information. It was established based on the higher PSNR, $\beta$, and EPF that the proposed network is more effective at removing speckle noise than multi-B-scans averaging while pre-serving spatial feature information better. Additionally, while the B-scan averaging method is still powerful in reducing speckle noise, this method can only be applied to very stable scanning methods in which adjacent frames are very similar, i.e. spatial differences are smaller than the lateral resolution of OCT and free from motion artifacts. Therefore, the B-scan averaging method cannot be applied when the frame interval is large or imaging is not stable due to motion. Of note, our method only uses a single B-scan image. Therefore, the proposed method can be applied to more diverse scanning situations, including intravascular OCT, in which helical scanning is applied and adjacent frames exhibit different features due to the relatively large frame interval. Fur-thermore, we demonstrate that the proposed method can enhance the resolution even for weak signals by comparing with contrast-adjusted images (Supplementary Fig. 3). We expect the proposed method to be useful when substantially enhanced OCT imaging performance is required.

Another unique feature of our method is its ability to enhance the spatial resolution without using super-resolution ground truth images, which cannot be obtained. Instead, we used the currently optimized OCT fringes/images and degraded OCT fringes/images as ground truth and input, respectively, to train the network. Because degraded OCT fringes/images were generated by a series of processes that mimic physical limitations or practical issues of

OCT, such as imperfect dispersion compensation and bandwidth truncation, the network learned to overcome these limitations. Subsequently, when the currently optimized OCT fringes/images were input to the network, enhanced OCT images were generated by the trained deep learning network.

Interestingly, enhancements of resolution and reduction of noise are clearly observed in both the state-of-the-art OCT images obtained from the same types of samples referenced during the training and from different types of samples from other OCT systems, demonstrating versatility and expandability. As a result, we present that this approach of directly accessing the fringes enhances the acquired OCT signal, enabling further resolution enhancement and noise reduction. Future work will include the subjective evaluation by clinicians to examine whether the pro-posed method is practically helpful in diagnosis or interpretation. This method can be applied to any Fourier domain OCT from which spectrograms can be obtained, such as swept-source OCT and spectral-domain OCT. We anticipate that the proposed deep learning-based OCT will contribute to broadening OCT usage in clinical and preclinical applications by providing images with higher resolution and SNR.

## Methods

**Data acquisition.** Data were collected using a customized benchtop-based SS-OCT[38]. with galvanometer scanners and a scan lens (LSM03, Thorlabs. Inc.) having axial and lateral resolution of 10 μm (air) and 13 μm, respectively. The OCT system has a central wavelength of 1290 nm, a bandwidth of 110 nm, an average output power of 40 mW, and a frame rate of 117 frames/s. The acquired training data consists of a total of 12 samples (5 different thyroid tissue specimens, finger nails, fingertip, cucumber, grape, lemon, pork meat, and Scoth tape). For each sample, a total of 5 sets were obtained in different regions. Each set consists of 1000 B-scans consisting of 1024 A-scans with a depth of 2048 pixels; thus, the total number of B-scans in the dataset is 60,000. The training set and the test set were constructed by dividing the pullback sets for each sample into a ratio of 8 to 2, resulting in 48,000 and 12,000 B-scans, respectively. The thyroid tissue specimen imaging was reviewed and exempted from deliberation by the Institutional Review Board of Gil medical center (GBIRB2021-241).

To investigate the expandable performance of the proposed deep learning approach, datasets not referenced during training were additionally acquired. With the same system as before, OCT images of droplets with 3 μm $TiO_2$ microspheres

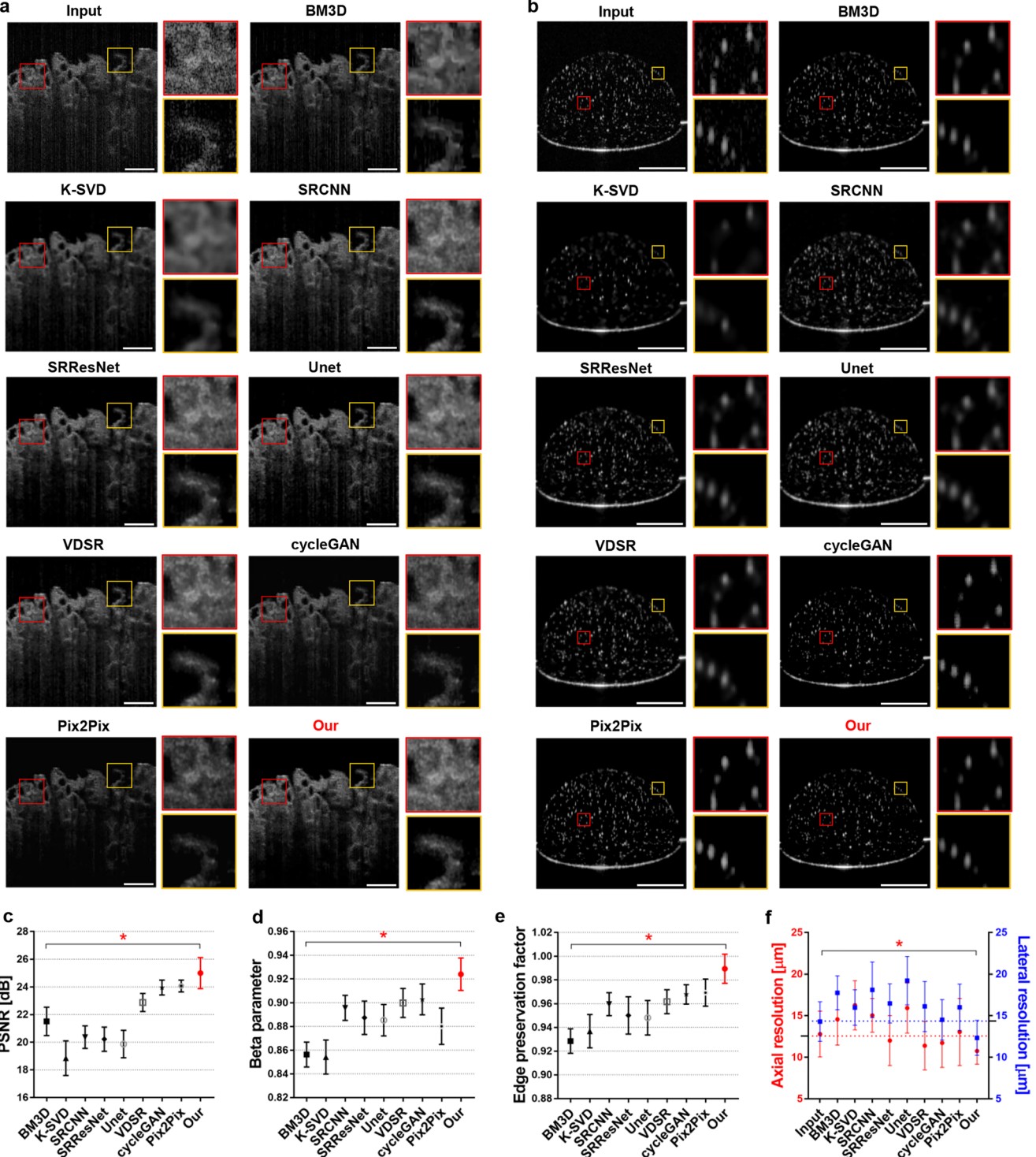

**Fig. 5 Results of comparative studies with other methods.** Example results for **a** thyroid tissue specimen and **b** TiO2 microsphere data. Input, BM3D, K-SVD, SRCNN, SRResNet, Unet, VDSR, cycleGAN, Pix2Pix, and our result are presented in order on both **a** and **b**. The input is an image reconstructed by the currently optimized OCT processing. The ROIs (red and yellow boxes) on the right sides of the images show magnified views (3X for **a**; 5X for **b**) from the same location. Scale bars, 1 mm. Metrics based on thyroid tissue specimen sample for quantitative comparison, **c** PSNR, **d** $\beta$, and **e** EPF. Our method showed superiority in terms of SNR and in preserving spatial features. **f** Axial and lateral resolution measured by microsphere data. Dashed lines represent the axial and lateral resolutions of input. While axial and lateral resolution enhancements were revealed in our method, the quantitative results of other methods were mostly inconspicuous for resolution enhancement. Error bars in **c**–**f** represent standard deviations. All of these statistical results were calculated for 200 randomly selected images, and the results of **f** was estimated from a total of 800 microsphere data. All multiple comparison results for our method were statistically significant ($P < 0.0001$ (*) according to a one-way ANOVA test).

(10086A, TSI Corp., USA) were acquired. In addition, arterial data were collected using a previously reported catheter-based SS-OCT[4,39], with an axial resolution of 11 μm (air) and a lateral resolution of 21 μm. The OCT system has a central wavelength of 1294 nm, a bandwidth of 110 nm, an average output power of 25 mW, and a frame rate of 114 frames/s. More details on the OCT system can be found in the previous works[4,39]. The acquired data consists of a pullback sets, some from in vivo swine coronary arteries implanted with bioresorbable scaffolds (a male Yucatan minipigs weighting ~15–20 kg, n = 1, Optipharm, Korea); others from rabbit abdominal aortas with atherosclerotic plaque (a male New Zealand white rabbit weighting ~3–3.5 kg, n = 1, DooYeol Biotech, Korea). Note that the OCT images helically scanned in polar coordinates were provided as inputs of the models, and the output images after the deep learning process were subjected to Cartesian transformation for visualization. All animal experiments were approved by the Institutional Animal Care and Use Committee of Korea University (KOREA-2019-0152-C1, KOREA-2021-0076) and were performed in accordance with national and institutional guidelines.

**Preparation of training datasets for NetA.** Training datasets for each model were separately generated by directly processing the raw interference fringes. The input and ground truth of the dataset for NetA were constructed by pairing OCT A-scans and interferograms with degraded axial resolution and the currently optimized OCT A-scans, respectively. Here, the currently optimized OCT A-scans are the A-scans with the best axial resolution achievable using our OCT system, which applies the best post-processing methods including background removal, k-line-arization, and numerical dispersion compensation[7]. On the other hand, the input was processed by applying imperfect numerical dispersion compensation and bandwidth truncation. Imperfect compensation in terms of dispersion was applied by randomly adjusting the polynomial fitting coefficients of dispersion, resulting in degraded axial resolution. The degree of degradation has been determined empirically to deteriorate the axial resolution by up to a factor of 2; detailed procedures can be found in the Supplementary Note 2. In addition, the axial resolution is inversely proportional to the bandwidth of the laser[7]. Accordingly, by truncating the bandwidth of the laser source, A-scans with degraded axial resolution can be obtained. The degree of bandwidth truncation was randomly selected within 0.5 times by applying Gaussian windows to the raw fringe before applying FFT. Example results of optimal and degraded OCT images, A-scans, and spectrograms for one of those A-scans are shown in Supplementary Fig. 6. The window and overlap size of the STFT were empirically set at 325 and 275 $cm^{-1}$, respectively, to generate a spectrogram with 14 different depth profiles derived from different spectral bands.

Each piece of data was normalized to improve the training efficiency of a gradient descent algorithm. Specifically, the transform results of STFT and FFT are a log compressed amplitude spectrum with a single precision floating point. Therefore, the range of the valid intensity range of the STFT and FFT results was normalized to between -1 and 1, respectively. After preprocessing, to augment the training dataset, two adjacent A-scans were randomly selected in each B-scan and randomly flipped in the horizontal direction.

**Training strategy of NetA.** The schematic of NetA architecture is shown in Fig. 6a; details are provided in the Supplementary Note 3, along with Supplementary Table 1. In the training phase, the generator receives degraded interferograms and A-scans and learns to generate paired optimal A-scans. Then, the reconstructed A-scans are fed to a discriminator that learns to discriminate between the ground truth A-scans and the generated A-scans, and then returns feedback (0 or 1) to the generator. As the generator and discriminator networks are, alternately, trained, the two networks compete to the theoretical limit at which the generated A-scan and the ground truth cannot be distinguished. The loss functions of the generator and the discriminator were defined as follows. The generator was trained with L1 loss, L2 loss, and gradient loss regarding only the axial direction, each defined as follows:

$$L1 = \frac{1}{m} \sum_{i=0}^{m-1} \left| A_{out}(i) - A_{groundtruth}(i) \right| \tag{1}$$

$$L2 = \frac{1}{m} \sum_{i=0}^{m-1} \left[ A_{out}(i) - A_{groundtruth}(i) \right]^2 \tag{2}$$

$$L_{gradient} = \frac{1}{m} \sum_{i=0}^{m-1} \left| \frac{\partial A_{out}}{\partial z} - \frac{\partial A_{real}}{\partial z} \right| \tag{3}$$

where m, $A_{out}$, and $A_{ground truth}$ are the sizes of the A-scans (2048 in our study), reconstructed A-scans, and ground truth A-cans, respectively. The adversarial loss included in the generator loss and the discriminator loss are defined as binary cross entropy (BCE), represented as follows:

$$BCE(x, y) = -\frac{1}{N} \sum_{i=1}^{N} (y \log x + (1-y) \log(1-x)) \tag{4}$$

where N, x, and y are the total number of outputs, the discriminator's result, and the actual label. For example, in the case of the ground truth, the actual label is 1, and the closer the discriminator's result is to 1, the smaller the loss. Using these

functions, the losses of the generator and the discriminator are defined as follows:

$$Loss_G = \lambda_1 L1 + \lambda_2 L2 + \lambda_3 L_{gradient} + \lambda_4 BCE(D(G(z)), 1) \tag{5}$$

$$Loss_D = BCE(D(z), 1) + BCE(D(G(z)), 0) \tag{6}$$

where λ is weight of each term, and D(·) and G(·) refer to the outputs of the discriminator and the generator, respectively. In $Loss_G$, the four λs are empirically defined as 1, 0.6, 0.9, and $10^{-4}$, respectively. The reason why $BCE(D(G(z)), 1)$ is included in $Loss_G$ is to use feedback to train it adversarially to determine whether the generator can deceive the discriminator well.

**Preparation of training datasets for NetB.** The dataset for training NetB was generated from the same raw data utilized for NetA, but with different pre-processing. The ground truth data were generated by B-scan averaging of 7 adjacent OCT images with the best compensation applied. By defining the frame interval of the averaged OCT images to be lower than the lateral resolution of the OCT system, proper noise reduction was achieved while preventing excessive spatial blurring. The interval between each frame is ~2 μm. Gaussian weights were also taken and averaged over the B-scans to retain as much spatial information as possible while suppressing speckle noise. B-scan images with degraded lateral resolution and prominent noise were generated and used as input data, after three processes of bandwidth truncation, lateral Gaussian filtering, and SNR deterioration. Note that the input B-scan is a single image in the middle of the OCT B-scans used to create the averaged image. Bandwidth truncation larger than 0.8 times the normal bandwidth was applied to introduce variation in the noise pattern with a slight degradation of axial resolution. After reconstructing the B-scan OCT images, lateral Gaussian filtering and SNR deterioration were performed. The filter size and sigma of Gaussian filtering were randomly selected within 5–15 and 1–5, respectively, to blur the B-scan without oversmoothing. Finally, SNR deterioration was applied either by lowering the intensity level to a maximum of 3 dB or by amplifying the noise level to a maximum of 3 dB. These processes were applied randomly. Each pair of input and ground truth data was normalized to a scale between −1 and 1.

**Training strategy of NetB.** The schematic of NetB architecture is shown in Fig. 6b; the details are summarized in the Supplementary Information, along with Supplementary Table 2. In the training phase, the generator is trained to reconstruct the desired output, B-scan averaged-optimal OCT image, by receiving the degraded OCT B-scan. The reconstructed B-scan is then fed to the discriminator to learn to discriminate the generated B-scan with a single output of 0 and 1, and then the feedback is returned to the generator. To train NetB, the loss functions are separately defined for generator and discriminator. Because the generators processes images, their loss functions comprise metrics used to measure the image quality. First, L1 and L2 losses were used to compare the difference between output and ground truth in pixels. Each loss is defined as follows:

$$L1 = \frac{1}{h} \frac{1}{w} \sum_{j=0}^{h-1} \sum_{i=0}^{w-1} \left[ I_{out}(i,j) - I_{groundtruth}(i,j) \right] \tag{7}$$

$$L2 = \frac{1}{h} \frac{1}{w} \sum_{j=0}^{h-1} \sum_{i=0}^{w-1} \left[ I_{out}(i,j) - I_{groundtruth}(i,j) \right]^2 \tag{8}$$

where h, w, $I_{out}$, and $I_{ground truth}$ are the height of the images, the width of the images, generated images, and ground truth images, respectively. The two losses above directly compare pixel-wise differences. However, it has been reported that using only these losses can blur the results[55]. Therefore, MS-SSIM loss was additionally adopted to ensure that structural features are well preserved; this process is defined as:

$$L_{MS-SSIM} = \frac{1 - MS SSIM(I_{out}, I_{groundtruth})}{2}. \tag{9}$$

The gradient loss is also utilized to compare the difference in gradient and variation in the height and width directions. By adding the gradient loss, the morphological edge features and texture information can be well preserved. The gradient loss is defined as follows:

$$L_{gradient} = abs\left( \frac{\partial I_{out}}{\partial x} - \frac{\partial I_{groundtruth}}{\partial x} \right) + abs\left( \frac{\partial I_{out}}{\partial z} - \frac{\partial I_{groundtruth}}{\partial z} \right) \tag{10}$$

where the operators ∂/∂x and ∂/∂z refer to directional intensity variations in the x and z directions. Using these loss functions, the losses of the generator and the discriminator are defined as follows:

$$Loss_G = \lambda_1 L1 + \lambda_2 L2 + \lambda_3 L_{MS-SSIM} + \lambda_4 L_{gradient} + \lambda_5 BCE(D(G(z)), 1) \tag{11}$$

$$Loss_D = BCE(D(z), 1) + BCE(D(G(z)), 0) \tag{12}$$

where λ is the empirically determined weight of each term, and $Loss_D$ is defined in the same way as in the A-model. In $Loss_G$, the five λs are empirically determined to be 0.8, 0.6, 1, 0.9, and $10^{-4}$, respectively. Details of the system implementation for training both networks are summarized in Supplementary Note 4.

**Fig. 6 Network architectures for NetA and NetB. a** The generator and discriminator of NetA consist of 13 blocks (comprising 4 Residual-in-Residual Dense Blocks (RRDB) and 1 skip connection block) and 11 blocks (comprising convolutional layers), respectively. **b** The generator and discriminator of NetB consist of 12 blocks (comprising 8 Residual-in-Residual Dense Blocks) and 11 blocks (comprising convolutional layers), respectively. Each block includes a 2D batch normalization and parametric rectified linear unit (PReLU) activation function. The block in the middle of the figure represents RRDB.

**Statistics and reproducibility**. The number of data and the statistical analysis are described in figure legends. All statistical analyses were performed using GraphPad Prism (GraphPad Prism 7.0, Graph Pad software Inc).

**Reporting summary**. Further information on research design is available in the Nature Portfolio Reporting Summary linked to this article.

## Data availability
All the source data presented in the main figures are available as Supplementary Data 1. The imaging datasets generated and/or analyzed during the current study are available from the corresponding author on reasonable request.

## Code availability
Deep learning-based OCT signal processing framework can be found in the oct-enhancement-framework repository (https://github.com/KAIST-BOOM/oct-enhancement-framework)

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

## Acknowledgements

This work was supported by the National Research Foundation of Korea (NRF) funded by the Ministry of Education, Science and Technology (NRF-2019M3A9E2066880 and RS-2023-00208888), and by a Korea Medical Device Development Fund grant funded by the Korean government (Ministry of Science and Information and Communication Technologies, Ministry of Trade, Industry and Energy, Ministry of Health and Welfare, Ministry of Food and Drug Safety) (1711138039, KMDF_PR_20200901_0054).

## Author contributions

W.L., H.S.N., and H.Y. conceived the concept of this study and contributed to the algorithms. H.S.N., J.Y.S., W.Y.O., and J.W.K. performed the OCT experiments and contributed to the acquisition of OCT data. W.L. drafted the manuscript; all authors contributed to the manuscript editing. H.Y. handled funding and supervision. All authors discussed the results and commented on the manuscript.

## Competing interests

The authors declare that they have no competing interests.
