## [Peer Review File · Communications Biology]

Reviewers' comments:

Reviewer #1 (Remarks to the Author):

When composing your report, the following questions might assist you in writing an incisive, well-justified review.

1. What are the major claims of the paper?

This is a very nice paper which presents a supervised learning method (using CNNs) to enhance the quality of OCT images (denoise/reduce speckle). It works on the raw interference data and uses a two-stage enhancement with two separate models which work on the A-scan (spectograms) and B-scan (lateral-resolution) modes of the data separately. The models are based on GANs and use the optimal A-scans (B-scans) as GT against degraded inputs passed through the networks.

2. Are they novel and will they be of interest to others in the community and the wider field? If the conclusions are not original, it would be helpful if you could provide relevant references.

Yes, the method seems novel in that it operates on the raw interference data and uses GANs. I believe the ideas might be limited to OCT and similar applications. However, the approach is well thought through and robustly applied.

3. Is the work convincing, and if not, what further evidence would be required to strengthen the conclusions?

The work is quite convincing. The quantitative results show that each stage (NetA and NetB) achieve MSE improvements on the input validation data.

The qualitative improvements are quite subtle and difficult to see. It would have been good to know whether clinicians found the enhancements useful as ultimately, I guess, it is them who would wish to see better reconstructions from the OCT. In my experience, practitioners may prefer more noisy images (rather than 'blurring' as noted by the authors which is the effect of some of the compared methods). To what extent the reduction in speckle is "better" for seeing features in the data is hard to gauge. In that sense, the in-vitro data examples in Figure 3 (and 4) are perhaps more convincing to a non-expert.

4. On a more subjective note, do you feel that the paper will influence thinking in the field?

There are already several methods existing which use deep learning for the same task (these are in fact compared in the paper, refs 45, 46, 47, 48 and discussed in the "comparative studies" section.

The presented methods are demonstrated to out-perform these and the methods based on statistical filtering, Figure 4, using PSNR, Beta parameter, Edge preservation factor, Axial resolution and lateral resolution. This is very good.

However, as OCT is not my area of expertise, I don't know to what extent these performance improvements and the two-network method would impact the field.

5. Please feel free to raise any further questions and concerns about the paper.

As a non-expert in OCT, I felt that there is scope to better introduce (briefly) the principles of OCT.

For example, I had to look outside the paper to fully appreciate the meaning of A and B scans and the concepts of depth and lateral resolution and how a standard OCT system acquires this information.

6. We would also be grateful if you could comment on the appropriateness and validity of any statistical analysis, as well the ability of a researcher to reproduce the work, given the level of detail provided.

The training/test data consisted of in-vivo scans from 35 'pullback sets', using a 80:20 split for training validation of the models on 14000 B-scans. The generalisation of performance was tested on in-vitro data, so testing was not done on in-vivo data. Otherwise, the statistical methods seem appropriate and well-conducted.

It may be good to perform multi-fold analysis and test several models on the same data, and then average the results.

A question for the authors would be to what extent are the B-scans are correlated when the probes are pulled back as the frame rate is 114 frames/s. It would be important then to make sure that adjacent scans did not end up in both training and validation sets which might results in over-fitting. One simple idea would be to compare a model trained on swine on rabbit data, and vice-versa.

I note that you show the dispersion compensation images in S7. I wonder if you could generate quantitative data of the enhancement performance against the degradation to see if the enhancement is linear and to what extent the chosen degradation for the training data affects the model generalistaion for different degrees of degradations. For example, if the degradation is small, does a model trained with higher degraded data over-smooth the output?

In terms of reproducibility, it would be useful for the authors to (1) publish some or all of their data; (2) make available their experimental codes which generate their results/and or the parameters of the trained models. This is becoming the standard practice in computational imaging, and would be beneficial for the wider community.

Reviewer #2 (Remarks to the Author):

The paper 'Deep learning-based image enhancement in optical coherence tomography by exploiting interference fringe'

The authors propose a deep learning-based OCT image enhancement framework that exploits raw interference fringes to achieve further enhancement from currently obtainable optimized images. The proposed framework for enhancing spatial resolution and reducing speckle noise in OCT images consists of two separate models: an A-scan-based network (NetA) and a B-scan-based network (NetB) based on GANs. They assessed the performance of the proposed method visually and quantitatively.

The qualitative comparison is outstanding but the quantitative is not clear. For example, the authors compare the proposed methods against the references 32-35?

The novelty is limited and my main concern is related to that authors should include other state-of-the-art methods

such as: Cycle-GAN or Pix-to-pix

For example the following papers:

<https://ieeexplore.ieee.org/abstract/document/9492218>

<https://opg.optica.org/josaa/abstract.cfm?uri=josaa-39-2-A62>

<https://www.spiedigitallibrary.org/conference-proceedings-of-spie/11313/1131309/Adversarial->

domain-adaptation-for-multi-device-retinal-OCT-segmentation/10.1117/12.2549839.short?SSO=1
<https://www.spiedigitallibrary.org/conference-proceedings-of-spie/12033/120333H/Device-specific-SD-OCT-retinal-layer-segmentation-using-cycle-generative/10.1117/12.2613066.short>

Some typos:

- iamges

Response to Reviewer 1's Comments

Comment 1: *This is a very nice paper which presents a supervised learning method (using CNNs) to enhance the quality of OCT images (denoise/reduce speckle). It works on the raw interference data and uses a two-stage enhancement with two separate models which work on the A-scan (spectrograms) and B-scan (lateral-resolution) modes of the data separately. The models are based on GANs and use the optimal A-scans (B-scans) as GT against degraded inputs passed through the networks.*

Our response: The authors would like to thank the reviewer for the thoughtful comments. We have provided point-by-point responses to each comment the reviewer has made on our manuscript. We believe that our responses to the reviewer's detailed comments have greatly improved our work.

Comment 2: *Yes, the method seems novel in that it operates on the raw interference data and uses GANs. I believe the ideas might be limited to OCT and similar applications. However, the approach is well thought through and robustly applied.*

Our response: We appreciate your insightful comment. We suggested the deep learning method that uses the spectrogram to more effectively utilize the information contained in the raw interference signal. We attempted to demonstrate that this approach is not limited to specific OCT systems and samples. In particular, this part was more firmly substantiated by using newly acquired dataset from more diverse samples in the revised manuscript.

Results (page 5, line 25)

The training dataset was constructed using a customized benchtop-based swept-source OCT (SS-OCT)³⁸ system from a variety of samples including thyroid tissue specimens, finger nails, fingertips, cucumbers, grapes, lemons, pork meat, and Scotch tape. Additional data not shown during training were also acquired with the same OCT system to demonstrate expandability. Furthermore, in vivo data from swine coronary artery and rabbit abdominal aorta, obtained using a customized catheter-based SS-OCT system^{4,39}, were also used to support more robust expandability of the proposed method.

Comment 3: *The work is quite convincing. The quantitative results show that each stage (NetA and NetB) achieve MSE improvements on the input validation data.*

The qualitative improvements are quite subtle and difficult to see. It would have been good to know whether clinicians found the enhancements useful as ultimately, I guess, it is them who would wish to see better reconstructions from the OCT. In my experience, practitioners may prefer more noisy images (rather than 'blurring' as noted by the authors which is the effect of some of the compared methods). To what extent the reduction in speckle is "better" for seeing features in the data is hard to gauge. In that sense, the in-vitro data examples in Figure 3 (and 4) are perhaps more convincing to a non-expert.

Our response: We thank the reviewer for the valuable comment. We fully agree with the reviewer's comment that obtaining evaluation from clinicians will help to more practically assess image quality performance. Although two clinicians (J.Y.S. and J.W.K) among the authors found the enhancements provided by the presented method to be significant and potentially useful for diagnosis, however, assessing the overall quality of images, such as noise reduction and resolution improvement, can be subjective. Therefore, it is challenging to obtain an objective and statistically meaningful evaluation from clinicians, and this will be our future work. A discussion on this issue has also been included in the Discussion section of the revised manuscript. To evaluate image quality as objectively as possible, we performed quantitative evaluation utilizing various image evaluation metrics. In addition, to signify the qualitative improvements, we further improved our method by adding more data from diverse samples and utilizing frame-averaged OCT images as output for network training. This improvement further reduced speckle noise, making the qualitative improvements more apparent. Quantitative improvements compared to other previous and state-of-the-art studies also support our findings. As noted by the reviewer, it is anticipated that these issues will be further resolved by reporting clinician's evaluation in future work on specific applications.

Discussion (page 10, line 10)

Future work will include the subjective evaluation by clinicians to examine whether the proposed method is practically helpful in diagnosis or interpretation.

Comment 4: *There are already several methods existing which use deep learning for the same task (these are in fact compared in the paper, refs 45, 46, 47, 48 and discussed in the "comparative studies" section.*

The presented methods are demonstrated to out-perform these and the methods based on statistical filtering, Figure 4, using PSNR, Beta parameter, Edge preservation factor, Axial resolution and lateral resolution. This is very good.

However, as OCT is not my area of expertise, I don't know to what extent these performance improvements and the two-network method would impact the field.

Our response: We thank the reviewer for the kind comment. Through comparative studies, we investigated whether our method that directly exploits raw interference signals could achieve more reliable performance. There have been no prior research employing the spectrogram information of raw interference signal as a direct deep learning input for this purpose. The performance of our method was supported by quantitative evaluations performed using various metrics and spatial resolution measurements. Therefore, we can expect that our approach can have performance advantages over other techniques that only process gray-scale OCT images. On the other hand, since the spectral bandwidth of the light source directly influences the spatial resolution and SNR from the perspective of image quality in OCT, the image quality can be improved by employing a light source with a broader spectral bandwidth. However, implementing hardware-based advancements, such as using expensive high-performance lasers, corresponding optics, and electronics, is challenging. Given its ability to enhance images to higher quality without changing any hardware, our technology is expected to have an impact in many fields utilizing OCT. Additionally, given the excellent results for samples not related to training as in **Fig.5b** of the revised manuscript (**Fig.4b** in the original manuscript), we can expect an advantage in versatility.

Results (page 8, line 13)

We compared our method with conventional methods based on statistical filtering (block-matching 3D (BM3D)¹⁵ and K-SVD⁴⁸) and six other previous deep learning techniques showing reliable performance in image improvement (super-resolution convolutional neural network (SRCNN)⁴⁹, super-resolution residual neural network (SRResNet)⁵⁰, Unet⁵¹, very-deep super-resolution (VDSR)⁵², cycle-consistent adversarial network (CycleGAN)⁵³, and paired image-to-image translation (Pix2Pix)⁵⁴). Deep learning techniques were adopted as-is for each of the proposed implementations, but retrained using the same dataset in this study. The training loss curves of each technique are shown in **Supplementary Fig.5**. **Fig.5** shows comparison results using datasets of thyroid carcinoma specimen (**Fig.5a**) and microspheres (**Fig.5b**).

Revised Figure 5

Figure 5 Results of comparative studies with other methods. Example results for **a** thyroid tissue specimen and **b** TiO₂ microsphere data. From the upper left areas in both **a** and **b**, input, BM3D, K-SVD, SRCNN, SRResNet, Unet, VDSR, cycleGAN, Pix2Pix, and our result are presented. The input is an image reconstructed by the currently optimized OCT processing. The ROIs (red and yellow boxes) on the right sides of the images show magnified views (3X for **a**; 5X for **b**) from the same location. Scale bars, 1 mm. Metrics based on thyroid tissue specimen sample for quantitative comparison, **c** PSNR, **d** β , and **e** EPF. Our method showed superiority in terms of SNR and in preserving spatial features. **f** Axial and lateral resolution measured by microsphere data. Dashed lines represent the axial and lateral resolutions of input. While axial and lateral resolution enhancements were revealed in our method, the quantitative results of other methods were mostly inconspicuous for resolution enhancement. All of these statistical results were calculated for 200 randomly selected images. All multiple comparison results for our method were statistically significant ($P < 0.0001$ (*) according to a one-way ANOVA test).

Comment 5: *As a non-expert in OCT, I felt that there is scope to better introduce (briefly) the principles of OCT. For example, I had to look outside the paper to fully appreciate the meaning of A and B scans and the concepts of depth and lateral resolution and how a standard OCT system acquires this information.*

Our response: We agree with the reviewer's suggestion, thus we have added the explanation of the basic principles of OCT to the Introduction of the revised manuscript.

Introduction (page 3, line 3)

OCT operates based on an interferometric technique that uses coherent detection of backscattered light from sample and reference arms using a broad-band light source². The difference in the optical length of each arm is encoded in the frequency domain of the detected interference signal. Therefore, the depth profile of a sample, commonly referred to as a A-scan, is typically retrieved by applying Fourier transform to the measured interference signal. The laser beam is then scanned laterally to obtain a two-dimensional cross-sectional OCT image with a depth-lateral axis, commonly referred to as a B-scan. Based on these fundamental principles, OCT with microscopic spatial resolution is widely used as a diagnostic tool in various medical fields, such as ophthalmology and cardiology^{3,4}. However, OCT applications often suffer from systemic limitations arising from the basic operating principle; these limitations include the presence of speckle noise, limited depth-of-focus (DOF), and degradation in spatial resolution. In detail, speckle noise deteriorates detailed morphological information by reducing contrast of OCT images⁵. The axial resolution is physically determined by the spectral bandwidth of the light source, while the lateral resolution is dominated mainly by the numerical aperture of the imaging optics⁶.

Comment 6-1: *The training/test data consisted of in-vivo scans from 35 'pullback sets', using a 80:20 split for training validation of the models on 14000 B-scans. The generalization of performance was tested on in-vitro data, so testing was not done on in-vivo data. Otherwise, the statistical methods seem appropriate and well-conducted.*

Our response: We appreciate the reviewer for the kind comment. In the revised manuscript, datasets from various samples were once again collected using the benchtop SS-OCT system. As a result, the training dataset consisted of in-vitro data only, while in-vivo intra-vascular data that were not referenced for training were additionally included in the test dataset. Therefore, the generalization performance of the proposed model, trained only using in-vitro data, could be tested and satisfactorily demonstrated on both in-vitro and in-vivo data.

Results (page 6, line 30)

In Fig.3a, an example of a cucumber cross-section shows that the speckle noise appearing within the tissue (arrowheads in Fig.3a) was reduced, while visual representation of structural features such as parenchyma (asterisks in Fig.3a) improved. Such improvement also appears in all of the samples referenced for training shown in Fig.3b-g. Fig.3h-j show examples of two other types of samples (microspheres and arterial cross-sections). Note that the data for these samples were used only for performance evaluation, not for training. In particular, the system used to image the arterial tissue was different from the one used to acquire the training data set. Since these samples have completely different structural features from those in the training set, the results can demonstrate the robust reliability and expandability of the proposed deep learning-based framework. Results of microspheres show enhanced spatial resolution, especially axial resolution, indicating significantly smaller bead sizes (red arrowheads in Fig.3h). Furthermore, the results for the arterial cross-section, shown in Fig.3i,j, confirm that our processing can achieve robust performance for biological samples obtained from other systems.

Comment 6-2: *It may be good to perform multi-fold analysis and test several models on the same data, and then average the results.*

Our response: We agree with the reviewer's opinion about the need for multi-fold analysis. To ensure that the proposed deep learning method was successfully trained without overfitting to particular samples, we performed additional training and validation on eight other datasets (thyroid tissue, fingertip, fingernail, cucumber, grape, lemon, pork meat, and Scotch tape). Each dataset was constructed by configuring one of the eight samples as a validation set and the remaining samples as a training set, as shown in **Supplementary Fig.4a**. The final image improvement performance was quantitatively assessed following the completion of training on each of the 8 datasets, as was the performance evaluation of the entire framework in the Result section of the manuscript. The evaluation results using the three metrics, PSNR, the beta parameter, and the edge preservation factor, are summarized in **Supplementary Fig.4b-d**. All metrics consequently presented comparable outcomes to those evaluated by originally trained models. This allowed us to support that our model was successfully trained without overfitting.

Results (page 8, line 5)

To confirm that both models were successfully trained without overfitting, a multi-fold analysis was also performed (**Supplementary Fig.4**). Of the eight datasets, one was used as a validation set and the other seven as training sets, which were used for further training and quantitative evaluation. Accordingly, all metrics showed equivalent results to those evaluated by the originally trained model, validating our model was successfully trained without overfitting.

Supplementary Information (page 14)
Added Supplementary Figure S4:

Figure S4 Performance evaluation result through multi-fold analysis. a Dataset configuration for multi-fold analysis. One of the eight samples served as the validation set, and others served as the training set, configuring up a total of eight datasets. **b-d** Quantitative evaluation results measured by **b** PSNR, **c** Beta parameter (β), and **d** EPF. The x-axis of each result displays the experiment number for each dataset as well as the original assessed result denoted by “our”. All metrics showed outcomes that are comparable to those determined by the values assessed by the original trained model. This demonstrates that our model was successfully trained without overfitting. In **b-d**, all multiple comparison results were statistically nonsignificant ($P > 0.999$ (ns) according to a one-way ANOVA test).

Comment 6-3: *A question for the authors would be to what extent are the B-scans are correlated when the probes are pulled back as the frame rate is 114 frames/s. It would be important then to make sure that adjacent scans did not end up in both training and validation sets which might results in over-fitting. One simple idea would be to compare a model trained on swine on rabbit data, and vice-a-versa.*

Our response: We thank the reviewer for the insightful comment. The arterial data consists of a helically scanned pullback sets. The correlation between adjacent B-scans of the acquired data was affected by the longitudinal pitch, which refers to the physical interval between B-scans. The SS-OCT system used in this study moves the catheter at a speed of 10 mm/s, while imaging is performed at a frame rate of 114 frames/s, resulting in an actual longitudinal pitch of about 88 μm . Therefore, while overall macroscopic structural similarity may appear, the correlation between two adjacent frames is low because the longitudinal pitch of the arterial data exceeds the system's lateral resolution. In fact, when new dataset was acquired for the revised manuscript, the interval between adjacent B-scans used for training and testing was also set above the lateral resolution of the system. Furthermore, we expect less concern about overfitting issues, as we acquired the new dataset from a variety of samples and re-evaluated performance on the arterial data that were not referenced for training. Additionally, we performed a multi-fold analysis, which showed that there were no issues with overfitting.

Materials and Methods (page 10, line 19)

Data were collected using a customized benchtop-based SS-OCT³⁸. with galvanometer scanners and a scan lens (LSM03, Thorlabs. Inc.) having axial and lateral resolution of 10 μm (air) and 13 μm , respectively. The OCT system has a central wavelength of 1,290 nm, a bandwidth of 110 nm, an average output power of 40 mW, and a frame rate of 117 frames/s. The acquired training data consists of a total of 12 samples (5 different thyroid tissue specimens, finger nails, fingertip, cucumber, grape, lemon, pork meat, and Scotch tape). For each sample, a total of 5 sets were obtained in different regions. Each set consists of 1,000 B-scans consisting of 1,024 A-scans with a depth of 2,048 pixels; thus, the total number of B-scans in the dataset is 60,000. The training set and the test set were constructed by dividing the pullback sets for each sample into a ratio of 8 to 2, resulting in 48,000 and 12,000 B-scans, respectively. The thyroid tissue specimen imaging was reviewed and exempted from deliberation by the Institutional Review Board of Gil medical center (GBIRB2021-241).

To investigate the expandable performance of the proposed deep learning approach,

datasets not referenced during training were additionally acquired. With the same system as before, OCT images of droplets with 3 μm TiO_2 microspheres (10086A, TSI Corp., USA) were acquired. In addition, arterial data were collected using a previously reported catheter-based SS-OCT^{4,39}, with an axial resolution of 11 μm (air) and a lateral resolution of 21 μm . The OCT system has a central wavelength of 1,294 nm, a bandwidth of 110 nm, an average output power of 25 mW, and a frame rate of 114 frames/s. More details on the OCT system can be found in the previous works^{4,39}. The acquired data consists of a pullback sets, some from in vivo swine coronary arteries implanted with bioresorbable scaffolds; others from rabbit abdominal aortas with atherosclerotic plaque. Note that the OCT images helically-scanned in polar coordinates were provided as inputs of the models, and the output images after the deep learning process were subjected to Cartesian transformation for visualization. All animal experiments were approved by the Institutional Animal Care and Use Committee of Korea University (KOREA-2019-0152-C1, KOREA-2021-0076) and were performed in accordance with national and institutional guidelines.

Comment 6-4: *I note that you show the dispersion compensation images in S7. I wonder if you could generate quantitative data of the enhancement performance against the degradation to see if the enhancement is linear and to what extent the chosen degradation for the training data affects the model generalization for different degrees of degradations. For example, if the degradation is small, does a model trained with higher degraded data over-smooth the output?*

Our response: We thank the reviewer for the valuable comment. In fact, severe image degradation completely destroys microscopic spatial information (at the resolution level of the system), and thus we assumed that deep learning would not be able to be trained to enhance such fine feature information. Therefore, we postulated that, in terms of performance and generalization of the deep learning model, it would be most appropriate to degrade the image quality to twice the spatial resolution. Additionally, degradation was performed randomly within the pre-determined range for different degrees of degradation to provide data augmentation and to improve generalization performance.

To substantiate our hypothesis and thoroughly address the reviewers' comment, we quantitatively assessed the improvement performance according to the level of degradation as shown in **Supplementary Fig.2**. Similar to the entire framework performance evaluation described in the manuscript, we employed the PSNR, β , and EPF metrics to quantify the degree of improvement achieved by our proposed method. In the original manuscript, we quantified the degree of improvement by evaluating the output against the currently optimized input, but this additional evaluation assesses the quality of the degraded input and output based on ground truth to provide a more comparative depiction of performance improvement in relation to the degree of degraded input. It should be noted that no quantitative evaluation was carried out on the input with a degree of degradation of 0.0, given that it is identical to the ground truth. The output results show that the image quality improves linearly with the degree of degradation present in the input data. The findings presented in the manuscript indicate that improved images can be obtained regardless of input degradation. Furthermore, even at higher levels of degradation, the results indicated that an image can be restored to a level of performance comparable to the original OCT image. These results demonstrate that the presented method has robust generalization performance across different levels of degradation, producing desirable results without over-smoothing, regardless of the degree of degradation.

Results (page 7, line 20)

The enhancement performance according to the degree of input degradation was additionally performed to thoroughly verify the generalization performance of the proposed method (Supplementary Fig.2). The findings indicate that the proposed method exhibits robust generalization performance for different levels of degradation, generating desirable outcomes while avoiding over-smoothing regardless of the degradation level.

Materials and Methods (page 11, line 20)

The degree of degradation has been determined empirically to deteriorate the axial resolution by up to a factor of 2; detailed procedures can be found in the Supplementary Information.

Supplementary Information (page 12)

Added Supplementary Figure S2:

Figure S2. Performance test of deep learning-based OCT image enhancement framework according to the degree of input data degradation. a PSNR, b β , and c EPF evaluation results of inputs and outputs according to the degree of degradation. In all graphs, the x-axis denotes the relative degree of degradation, where “0.0” corresponds to the input with no degradation, i.e., equivalent to the ground truth (currently optimized OCT data). This test evaluates the quality of the degraded input and corresponding output against ground truth, presenting a comparative representation of the extent of performance improvement against the degree of degradation. Note that quantitative evaluation is not conducted on the input when the degradation degree is “0.0” because it is the same as the ground truth. The results show that the presented method can produce enhanced images when the input is either undegraded (currently optimized OCT image) or when the level of degradation is low. The findings also demonstrate that, even images with higher levels of degradation can be restored to comparable performance levels to the original OCT image.

Comment 6-5: *In terms of reproducibility, it would be useful for the authors to (1) publish some or all of their data; (2) make available their experimental codes which generate their results/and or the parameters of the trained models. This is becoming the standard practice in computational imaging, and would be beneficial for the wider community.*

Our response: We thank the reviewer for the kind suggestion. We provided access to the training-related codes and data via Github.

Code availability (page 14, line 20)

Deep learning-based OCT signal processing framework can be found in the oct_enhancement_framework repository (<https://github.com/KAIST-BOOM/oct-enhancement-framework>)

Response to Reviewer 2's Comments

Comment 1: *The authors propose a deep learning-based OCT image enhancement framework that exploits raw interference fringes to achieve further enhancement from currently obtainable optimized images. The proposed framework for enhancing spatial resolution and reducing speckle noise in OCT images consists of two separate models: an A-scan-based network (NetA) and a B-scan-based network (NetB) based on GANs. They assessed the performance of the proposed method visually and quantitatively.*

The qualitative comparison is outstanding but the quantitative is not clear. For example, the authors compare the proposed methods against the references 32-35? The novelty is limited and my main concern is related to that authors should include other state-of-the-art methods such as: Cycle-GAN or Pix-to-pix. For example the following papers:

<https://ieeexplore.ieee.org/abstract/document/9492218>

<https://opg.optica.org/josaa/abstract.cfm?uri=josaa-39-2-A62>

<https://www.spiedigitallibrary.org/conference-proceedings-of-spie/11313/1131309/Adversarial-domain-adaptation-for-multi-device-retinal-OCT-segmentation/10.1117/12.2549839.short?SSO=1>

<https://www.spiedigitallibrary.org/conference-proceedings-of-spie/12033/120333H/Device-specific-SD-OCT-retinal-layer-segmentation-using-cycle-generative/10.1117/12.2613066.short>

Our response: We thank the reviewer for the insightful criticism. We fully concur with the reviewer's criticism that comparisons with state-of-the-art methods should be further investigated from the perspective of quantitative comparison. We first retrained our model using the new dataset to achieve further speckle noise reduction and resolution enhancement performance in the revised manuscript. In particular, the speckle noise reduction capability was specifically enhanced by employing the frame-averaged images of 7 adjacent optimal B-scan OCT images as the NetB's ground truth. We were also successful in revealing robust generalization performance for more diverse data. Based on these improvements, a new comparative study was performed using the recommended state-of-the-art methods (cycleGAN and Pix2Pix). Other previous studies that presented unique data collecting strategy were excluded because direct comparison with our method was difficult. The results of the new comparative study are summarized in **Fig.5** of the revised manuscript. The results were superior, as demonstrated in the quantitative comparison, even though the two state-of-the-art methods were better than the other deep learning techniques utilized in the previous comparative study.

These results support the hypothesis that our method of exploiting the raw interference signal of OCT is superior in terms of image enhancement. To incorporate the reviewer's suggestion, we also included recommended papers as references in the introduction of the revised manuscript.

Introduction (page 3, line 29)

Recent researches have shown that deep learning can outperform handcrafted feature descriptors in a number of imaging processing fields¹⁸⁻²³. Inspired by these studies, deep learning is being applied actively to various optical imaging modalities, including OCT, for super resolution and noise reduction²⁴⁻²⁹.

Results (page 5, line 20)

Degraded B-scan OCT images were used as input, and frame-averaged images of 7 adjacent optimal B-scan OCT images were used as ground truth for NetB (**Fig.1b**). Note that the total interval of the 7 OCT images was specified at the lateral resolution level of the OCT system to achieve adequate noise reduction while avoiding excessive spatial smoothing.

Results (page 5, line 25)

The training dataset was constructed using a customized benchtop-based swept-source OCT (SS-OCT)³⁸ system from a variety of samples including thyroid tissue specimens, finger nails, fingertips, cucumbers, grapes, lemons, pork meat, and Scotch tape. Additional data not shown during training were also acquired with the same OCT system to demonstrate expandability. Furthermore, in vivo data from swine coronary artery and rabbit abdominal aorta, obtained using a customized catheter-based SS-OCT system^{4,39}, were also used to support more robust expandability of the proposed method.

Results (page 6, line 30)

In **Fig.3a**, an example of a cucumber cross-section shows that the speckle noise appearing within the tissue (arrowheads in **Fig.3a**) was reduced, while visual representation of structural features such as parenchyma (asterisks in **Fig.3a**) improved. Such improvement also appears in all of the samples referenced for training shown in **Fig.3b-g**. **Fig.3h-j** show examples of two other types of samples (microspheres and arterial cross-sections). Note that the data for these samples were used only for performance evaluation, not for training. In

particular, the system used to image the arterial tissue was different from the one used to acquire the training data set. Since these samples have completely different structural features from those in the training set, the results can demonstrate the robust reliability and expandability of the proposed deep learning-based framework. Results of microspheres show enhanced spatial resolution, especially axial resolution, indicating significantly smaller bead sizes (red arrowheads in **Fig.3h**). Furthermore, the results for the arterial cross-section, shown in **Fig.3i,j**, confirm that our processing can achieve robust performance for biological samples obtained from other systems.

Results (page 8, line 13)

We compared our method with conventional methods based on statistical filtering (block-matching 3D (BM3D)¹⁵ and K-SVD⁴⁸) and six other previous deep learning techniques showing reliable performance in image improvement (super-resolution convolutional neural network (SRCNN)⁴⁹, super-resolution residual neural network (SRResNet)⁵⁰, Unet⁵¹, very-deep super-resolution (VDSR)⁵², cycle-consistent adversarial network (CycleGAN)⁵³, and paired image-to-image translation (Pix2Pix)⁵⁴). Deep learning techniques were adopted as-is for each of the proposed implementations, but retrained using the same dataset in this study. The training loss curves of each technique are shown in **Supplementary Fig.5**. **Fig.5** shows comparison results using datasets of thyroid carcinoma specimen (**Fig.5a**) and microspheres (**Fig.5b**).

Materials and Methods (page 13, line 5)

The ground truth data were generated by B-scan averaging of 7 adjacent OCT images with the best compensation applied. By defining the frame interval of the averaged OCT images to be lower than the lateral resolution of the OCT system, proper noise reduction was achieved while preventing excessive spatial blurring. The interval between each frame is approximately 2 μm . Gaussian weights were also taken and averaged over the B-scans to retain as much spatial information as possible while suppressing speckle noise.

Revised Figure 3

Figure 3 Blind testing performance of deep learning-based OCT image enhancement framework. Left and right columns represent currently optimized input and enhanced final output, respectively. Note that, in these results, the currently optimized OCT data were the input. Final output is the result of sequentially applying both NetA and NetB to the input. Each figure represents **a** cucumber, **b** grape, **c,d** thyroid tissue specimen, **e** finger nail, **f** Scotch tape, **g** pork meat, **h** droplet with TiO₂ microspheres, and **i,j** arterial cross-section. The ROIs (red and yellow boxes) on the right side of the image show magnified views (3X for **a-g** and **i-j**; 5X for **h**). Scale bars, 1 mm.

Revised Figure 5

Figure 5 Results of comparative studies with other methods. Example results for **a** thyroid tissue specimen and **b** TiO₂ microsphere data. From the upper left areas in both **a** and **b**, input, BM3D, K-SVD, SRCNN, SRResNet, Unet, VDSR, cycleGAN, Pix2Pix, and our result are presented. The input is an image reconstructed by the currently optimized OCT processing. The ROIs (red and yellow boxes) on the right sides of the images show magnified views (3X for **a**; 5X for **b**) from the same location. Scale bars, 1 mm. Metrics based on thyroid tissue specimen sample for quantitative comparison, **c** PSNR, **d** β , and **e** EPF. Our method showed superiority in terms of SNR and in preserving spatial features. **f** Axial and lateral resolution measured by microsphere data. Dashed lines represent the axial and lateral resolutions of input. While axial and lateral resolution enhancements were revealed in our method, the quantitative results of other methods were mostly inconspicuous for resolution

enhancement. All of these statistical results were calculated for 200 randomly selected images. All multiple comparison results for our method were statistically significant ($P < 0.0001$ (*) according to a one-way ANOVA test).

Supplementary Information (page 15)

Revised Supplementary Figure S5:

Figure S5. Training loss curves of other deep learning techniques for comparative studies. This figure represents training loss plots of deep learning techniques (SRCNN, SRResNet, Unet, VDSR, cycleGAN, and Pix2Pix) retrained for comparative studies. The loss function of each techniques was adopted as in the literature, and each plot was normalized. Losses gradually converged for all techniques, indicating successful training.

REVIEWERS' COMMENTS:

Reviewer #1 (Remarks to the Author):

I believe the authors have done a reasonable job of addressing my original review comments.

I think the revised manuscript has been improved by this and the results are move convincing.

I don't have any additional suggestions or comments on the revision.